# Development of an SI-traceable transmission curve reference material to improve quantitation and comparability of PTR-MS measurements

David R. Worton[1], Sergi Moreno[1,#], Kieran O'Daly[1], Rupert Holzinger[2]

[1] National Physical Laboratory, Hampton Road, Teddington, TW11 0LW, United Kingdom

[2] Institute for Marine and Atmospheric Research, IMAU, Utrecht University, Utrecht, the Netherlands

*Correspondence to*: David R. Worton (dave.worton@npl.co.uk)

[#] Now at World Meteorological Organization (WMO), 7bis Avenue de la Paix, C.P. 2300

CH-1211, Geneva 2, Switzerland

**Abstract.** Since its inception more than two decades ago proton-transfer-reaction mass-spectrometry (PTR-MS) has established itself as a powerful technique for the measurements of a wide range of volatile organic compounds (VOCs) with high time resolution and low detection limits and without the need for any sample pre-treatment. As this technology has matured and its application become more widespread there is a growing need for accurate and traceable calibration to ensure measurement comparability. As a result of the large number of VOCs detectable with PTR-MS it is impractical to have a calibration standard or standards that cover all observable compounds. However, recent work has demonstrated that quantitative measurements of uncalibrated compounds are possible provided that the transmission curve is accurately constrained. To enable this, a novel traceable multi-component gas reference material containing 20 compounds spanning a mass range of 32 to 671 has been developed. The development and compositional evolution of this reference material is described along with an evaluation of its accuracy and stability. This work demonstrates that for the majority of components the accuracy is < 5 % (most < 3 %; < 10 % for hexamethylcyclotrisiloxane (D3-siloxane) and 1,2,4-trichlorobenzene (1,2,4-TCB)) with stabilities of > 2 years (> 1 year for acetonitrile, methanol and perfluorotributylamine (PFTBA)).

## 1 Introduction

Proton-transfer-reaction mass-spectrometry (PTR-MS) is a technique that allows simultaneous measurements of multiple volatile organic compounds (VOCs) in real-time ($\leq 1$ s) with low detection limits (pmol mol$^{-1}$) and without any sample pre-treatment (Lindinger et al., 1998; Hansel et al., 1999). Most VOCs, with the exception of alkanes with less than five carbon atoms, have proton affinities larger than water ($691 \pm 3$ kJ mol$^{-1}$) and are therefore detectable with PTR-MS. Also, because most VOCs have proton affinities below 900 kJ mol$^{-1}$ there is minimal excess energy following proton transfer resulting in minimal fragmentation. For these reasons it is a very convenient measurement technique for a wide range of applications. Over the last two decades PTR-MS has become an important and widely applied tool for VOC measurements that has resulted in major advances in the field of atmospheric sciences (De Gouw and Warneke, 2007; Park et al., 2013; Yuan et al., 2017). It has also been applied in the medical sector for the detection of VOCs to diagnose diseases or disease states (Beauchamp et al., 2013) and in the food and beverage industry for characterising flavour and odour (Biasioli et al., 2011).

Multiple manufacturers now produce and commercially sell PTR-MS instruments globally that differ in the production and detection of ions including different types of mass spectrometer. Therefore, as its application becomes more widespread, and more datasets are generated there is an increasing need for accurate calibration and measurement comparability. Additionally, as part of the European funded Aerosols, Clouds and Trace Gases Infrastructure (ACTRIS) project (https://www.actris.eu/) there is an interest to establish PTR-MS as a technique for long-term monitoring of VOCs, which emphasises the need for a robust metrological infrastructure to control and assure the quality of data produced by monitoring stations performing these measurements. However, the lack of traceable reference materials to calibrate PTR-MS instruments presents challenges in the pursuit of obtaining comparable results and is an obstacle to long-term studies. Primary reference materials (PRMs) prepared by gravimetry in high pressure cylinders by national metrology institutes (NMIs) underpin the accuracy (trueness) and comparability of measurement data through traceability to the international system of units (SI). Traceability has been demonstrated as a critical component for chemical measurements that ensures the comparability, stability and coherence in measurements providing confidence in measurement results (Brown and Milton, 2007). PRMs produced by NMIs represent the highest point in the traceability chain and the accuracy and international comparability is ensured through key comparisons organised within the Consultative Committee on Amount of Substance Gas Analysis Working Group (CCQM-GAWG) and in regional comparisons organised within the Regional Metrology Organisations (RMOs), e.g., EURAMET (Europe).

As a result of the numerous VOCs detectable with PTR-MS it is impractical to have a calibration standard or standards that cover all observable compounds. However, since the conception of PTR-MS, there has been awareness for the potential of this technique to provide quantitative measurements for compounds without the need for specific calibration materials (Hansel et al., 1999). The basis for this is that the amount fraction of compound R ([R]) can be determined from (Taipale et al., 2008):

$$[R] = \frac{1}{k\Delta t} \times \frac{I(RH^+)}{T(RH^+)} \times \left(\frac{I(H_3O^+)}{T(H_3O^+)}\right)^{-1}$$ Eq. 1

Where k is the proton transfer reaction rate coefficient, $\Delta t$ is the reaction time, $I(RH^+)$ and $I(H_3O^+)$ are the observed ion count
rates for the protonated ion of compound R ($RH^+$) and the hydronium ion ($H_3O^+$), respectively. $T(RH^+)$ and $T(H_3O^+)$ are the
transmission efficiencies for $RH^+$ and $H_3O^+$ ions, respectively. The transmission coefficients are predominantly mass
dependent, but they can also vary in time (De Gouw et al., 2003; Ammann et al., 2004; Steinbacher et al., 2004). Proton transfer
reaction rate coefficients can be measured and/or predicted using quantum methods (Zhao and Zhang, 2004). If specific rate
coefficients are agreed within the community for specific compounds and are widely used this would negate the role of different
rate constants on measurement comparability (Table S1, Supporting Information). The reaction time and observed ion count
rates are all measured parameters leaving just the transmission coefficients as variables required for quantitative measurements
without specific calibrations. Cappellin et al. (2012) demonstrated the quantitative properties of one type of PTR-MS
instrument by assuming a theoretical transmission based on the duty cycle of the time-of-flight mass analyser. However, for
newer generation instruments that employ advanced ion optics to improve sensitivity, it is necessary to determine the mass-
dependent transmission experimentally as the transmission of the system diverges from theory at low masses. Deviations can
also occur at high masses due to poor tuning and/or ageing of the ion detection system (Müller et al., 2014).

There are several highly cited publications that explore best practices in PTR-MS measurements (e.g., Blake et al., 2009; De
Gouw and Warneke, 2007; Yuan et al., 2017), including methods to calibrate and retrieve the mass dependent transmission
(Taipale et al., 2008). However, many of these methods are slow and labour intensive and as a result calibrations and
transmission curve retrievals are not performed frequently enough. This has limited the application of PTR-MS to mostly short
campaign-scale intensive deployments and only a few groups have utilised PTR-MS for long-term studies (Holzinger et al.,
2006; Taipale et al., 2008). However, recent work by Holzinger et al., (2019) has demonstrated: (i) a new method to retrieve
the mass-dependent transmission from fast calibrations that should enable more frequent calibrations and (ii) the validity of a
simple reaction kinetics approach to quantify measurements of uncalibrated compounds from different PTR-MS instruments
with an accuracy of $\leq 30\ \%$ provided the transmission curve is accurately constrained. A prototype PRM, 0917a reported in
this work, was initially developed by the National Physical Laboratory (NPL) and employed for the PTR-MS intercomparison
campaign at the CESAR observatory in the central Netherlands (Holzinger et al., 2019). Following this comparison exercise
improvements to the composition were needed to include additional compounds in the mass-to-charge (m/Q) 150 – 400 Th
range to provide a more robust retrieval of the mass-dependent transmission.

In this paper, the development and compositional evolution of PRMs and certified reference materials (CRMs) specific for
constraining the PTR-MS transmission curve are described, including an evaluation of the accuracy through comparisons
validating the gravimetric preparation of various different PRMs of similar composition and an assessment of their long-term
stabilities. For details on how to use the RMs to constrain the PTR-MS transmission curve the reader is directed to H

## 91  2 Experimental methods

### 92  2.1 Gravimetric preparation of primary reference materials

The PRMs were prepared at four distinct timepoints (September 2017, December-January 2018, August 2019 and August
2021) and the compositions evolved over this timeframe (Table 1) due to improvements in the preparation and validation
techniques (e.g., 1,2,4-trichlorobenzene; 1,2,4-TCB) or due to requests from the PTRMS community for inclusion of new
components (e.g., dimethyl sulfide; DMS). All the PRMs were prepared gravimetrically in accordance with ISO 6142-1:2015
(ISO, 2015) from pure components. All pure components were purity analysed in accordance with ISO 19229 (ISO, 2019).
Table S2 (Supporting Information) provides the sources and purities for each component and shows that all chemicals with
the exception of perfluorotributylamine (PFTBA) were $\geq 98$ % pure. Table S2 (Supporting Information) also shows the boiling
points and vapour pressures for all compounds. All components were liquids at room temperature and pressure, with the
exception of propane (gas) and hexamethylcyclotrisiloxane (D3-siloxane; solid). As a solid under room temperature and
pressure conditions, D3-siloxane needed to be dissolved into a solvent to enable its addition to the cylinder. Further details are
given in the Supporting Information (Supplementary text: Preparation of D3-siloxane reference materials).

All PRMs were prepared in 10 L aluminium cylinders (Luxfer) with a proprietary passivation treatment (Spectraseal™, BOC)
and BS341 no. 15 outlet diaphragm valves (Ceodeux). Cylinders were evacuated using an oil free pump (Scrollvac SC15D,
Leybold Vacuum) and turbo molecular pump with magnetic bearing (Turbovac 340M, Leybold Vacuum) to a pressure of $< 3$
$\times 10^{-7}$ mbar. Individual compounds were added to the evacuated cylinder via a transfer vessel (capped $^1/_8$" diameter tube, with
a nominal volume of 1 mL, Swagelok, electro-polished stainless steel). The transfer vessel was weighed against a tare vessel
matched for size and shape before and after each addition into the evacuated cylinder (Mettler-Toledo XP2004S). The ultra-
high purity nitrogen balance gas (BIP$^+$, Air Products) was added via direct addition to the cylinder, through purged $^1/_{16}$" tubing
(Swagelok, electro-polished stainless steel). For the vast majority of compounds, they were initially produced as binary parent
mixtures at amount fractions $> 10$ µmol mol$^{-1}$ (typically at nominally 50 µmol mol$^{-1}$) though some were produced as ternary
or quaternary mixtures containing two or three compounds together in the same parent mixture. A full breakdown of the 50
parent mixtures used to prepare the six PRMs developed in this work are shown in Table 2. Aliquots of each of these parent
mixtures were added by direct addition to an evacuated cylinder to produce a final mixture containing all 16 to 20 VOCs at
nominal amount fractions of 1 µmol mol$^{-1}$ in a balance of nitrogen.

**Table 1.** Overview of composition (name, formula, CAS#), the protonated monoisotopic molecular ion and any major fragment ions, formed following protonation in $H_3O^+$ mode) for 20 compounds included in the PRM and CRM reference materials and the key comparisons through which the traceability to the international community is derived.

| Compound[a] | Formula | CAS # | m/Q [Th] protonated | fragments | Traceability (Reference, if applicable) |
|---|---|---|---|---|---|
| methanol | $CH_3OH$ | 67-56-1 | 33.033 | - | EURAMET-1305 OVOCs (Worton et al., 2022), CCQM-K174[b] |
| acetonitrile | $CH_3CN$ | 75-05-8 | 42.034 | - | CCQM-K174[b] |
| acetaldehyde | $CH_3CHO$ | 75-07-0 | 45.033 | - | CCQM-K174[b] |
| propane[c] | $C_3H_8$ | 74-98-6 | not detected | - | CCQM-K111 (Veen et al., 2017) |
| ethanol | $C_2H_5OH$ | 64-17-5 | 47.049 | - | CCQM-K93 (Brown et al., 2013), EURAMET-1305 OVOCs (Worton et al., 2022), CCQM-K174[b] |
| acetone | $CH_3OCH_3$ | 67-64-1 | 59.049 | - | EURAMET-1305 OVOCs (Worton et al., 2022), CCQM-K174[b] |
| DMS | $C_2H_6S$ | 75-18-3 | 63.026 | - | CCQM-K94 (Lee et al., 2016), CCQM-K165 (Lee et al., 2022) |
| isoprene | $C_5H_8$ | 78-79-5 | 69.070 | 41.039 | EURAMET-886 (Grenfell et al., 2008; Grenfell et al., 2010) |
| MVK | $C_4H_6O$ | 78-94-4 | 71.049 | - | CCQM-K174[b] |
| MEK | $C_4H_8O$ | 78-93-3 | 73.065 | - | CCQM-K174[b] |
| benzene | $C_6H_6$ | 71-43-2 | 79.054 | - | CCQM-K10.2018 (Cecelski et al., 2022) |
| toluene | $C_7H_8$ | 108-88-3 | 93.07 | - | CCQM-K10.2018 (Cecelski et al., 2022) |
| m-xylene | $C_8H_{10}$ | 108-38-3 | 107.086 | - | CCQM-K10.2018 (Cecelski et al., 2022) |
| 1,2,4-TMB | $C_9H_{12}$ | 95-63-6 | 121.101 | - | EURAMET-886 (Grenfell et al., 2008; Grenfell et al., 2010) |
| 1,2,4-TFB | $C_6H_3F_3$ | 367-23-7 | 133.026 | - | - |
| 3-carene | $C_{10}H_{15}$ | 13466-78-9 | 137.132 | 81.070 | CCQM-K121 (Liaskos et al., 2018) |
| 1,2,4-TCB | $C_6H_3Cl_3$ | 120-82-1 | 180.937 | - | - |
| D3-siloxane[d] | $C_6H_{18}O_3Si_3$ | 541-05-9 | 223.064 | 207.033, 225.033[e] | EURAMET 1305 Siloxanes[b] |
| D4-siloxane[d] | $C_8H_{24}O_4Si_4$ | 556-67-2 | 297.082 | 281.051, 283.030, 299.062[e] | EURAMET 1305 Siloxanes[b] |
| D5-siloxane[d] | $C_{10}H_{30}O_5Si_5$ | 541-02-6 | 371.101 | 355.070, 373.081[e] | EURAMET 1305 Siloxanes[b] |
| PFTBA | $C_{12}F_{27}N$ | 311-89-7 | not detected | 651.961, 413.977 | - |

[a]Short names shown here but preferred IUPAC names are: propan-2-one (acetone), (Methylsulfanyl)methane (dimethyl sulfide; DMS), 2-Methylbuta-1,3-diene (isoprene), but-3-en-2-one (methyl vinyl ketone; MVK), butan-2-one (methyl ethyl ketone; MEK), 1,2,4-trimethyl benzene (1,2,4-TMB), 1,2,4-trifluoro benzene (1,2,4-TFB), 3,7,7-trimethylbicyclo[4.1.0]hept-3-ene (3-carene), 1,2,4-trichlorocbenzene (1,2,4-TCB), hexamethylcyclotrisiloxane (D3-siloxane), octamethylcyclotetrasiloxane (D4-siloxane), decamethylcyclopentasiloxane (D5-siloxane), and perfluorotributylamine (PFTBA). [b]Comparison in progress at the time of publication. [c]Not detectable in PTRMS in $H_3O^+$ mode but included as an internal standard. [d]Further information on the mechanisms yielding product ions and fragments in Fig. S4 (Supporting Information). [e]Methyl/hydroxyl group switch.

**Table 2.** Composition, hierarchies and parent cylinder IDs (dates prepared) for all components for the six PRMs (0917a, 0917b, 1218, 0119, 0819 and 0821) prepared in this work. The PRMs are identified by the date and year of their preparation (MMYY). As the first two were produced at the same time the suffices a and b and added to distinguish them. The colour scheme in the table headers is matched to that used in the figures throughout the paper.

| Compound | Cylinder ID (Date Prepared) | | | | | |
|---|---|---|---|---|---|---|
| | 0917a (18/09/2017) | 0917b (18/09/2017) | 1218 (04/12/2018) | 0119 (02/01/2019) | 0819 (23/08/2019) | 0821 (31/08/2021) |
| methanol | | A463 (13/02/2015) | | | A410 (24/01/2013) | A602 (26/04/2018) |
| acetonitrile | | A389R (11/02/2015) | | | A403 (11/10/2012) | A670R (01/04/2021) |
| acetaldehyde | | A400R (02/02/2015) | | | A402 (11/10/2012) | 2832 (01/04/2021) |
| propane | | D910381R (18/11/2014) | | | NG561 (16/10/2014) | - |
| ethanol | | A463 (13/02/2015) | | | A410 (24/01/2013) | A602 (26/04/2018) |
| acetone | | A463 (13/02/2015) | | | VOC6 (05/05/2009) | A602 (26/04/2018) |
| DMS | | - | | 2106 (21/02/2017) | NG388 (13/09/2012) | 3073 (16/11/2020) |
| isoprene | | D292194R (13/01/2011) | | | VOC6 (05/05/2009) | D994138R2 (28/09/2020) |
| MVK | | 2064 (24/06/2016) | | | 2088 (24/06/2016) | 3070 (02/08/2021) |
| MEK | | A389R (11/02/2015) | | | A403 (11/10/2012) | 3070 (02/08/2021) |
| benzene | | D910381R (18/11/2014) | | | D842635R (13/10/2015) | D618317 (15/08/2018) |
| toluene | | | | - | | D600070 (19/03/2018) |
| m-xylene | D641688 (04/03/2010) | | | D618307 (19/03/2018) | D994171 (20/03/2013) | D618307 (19/03/2018) |
| 1,2,4-TMB | D442684 (02/03/2017) | | D711530 (26/11/2018) | D711532 (26/11/2018) | | D442684 (02/03/2017) |
| 1,2,4-TFB | | A569 (11/09/2017) | | | 2810 (21/08/2019) | D723197R (14/06/2021) |
| 3-carene | D090493 (18/11/2014) | | | D711532 (21/11/2018) | | |
| 1,2,4-TCB | - | | D641970R (31/07/2018) | | A568 (18/08/2017) | D723197R (14/06/2021) |
| D3-siloxane | - | - | 2586 (07/11/2018) | 2693 (07/11/2018) | 2586 (07/11/2018) | 3134 (22/06/2021) |
| D4-siloxane | A582 (15/08/2017) | A567 (18/08/2017) | - | | A582 (15/08/2017) | A629R (22/06/2021) |
| D5-siloxane | A582 (15/08/2017) | A567 (18/08/2017) | A629 (26/11/2018) | | A582 (15/08/2017) | A629R (22/06/2021) |
| PFTBA | | | - | | | D961497 (30/07/2021) |

## 2.2 Analytical methods

To perform the analytical validation a method was developed on a gas chromatograph (GC; Agilent Technologies 7890) instrument equipped with both a flame ionisation detector (FID) and electron ionisation (70 eV) mass spectrometer (Agilent 5975; GC-FID/MS system). Samples were introduced using a 6 port 2 position valve (VICI) and a fixed sampling loop (1 mL). The column effluent was split to both detectors simultaneously by using a detector splitter plate (Agilent Technologies). Separation was achieved for all components using a DB-624 capillary column (J&W; 75m × 0.53 mm, df = 3 µm). The carrier gas was helium and the flow was held constant at 5 mL min$^{-1}$, with a temperature program starting at 30 °C held for 10 minutes, ramped at 10 °C min$^{-1}$ to 120 °C and held for 15 minutes before a final ramp of 50 °C min$^{-1}$ was applied to a final temperature of 200 °C, which was held for a further 10 minutes. The total run time was 46 minutes.

Low FID responses for methanol and acetaldehyde presented some analytical challenges because the observed peaks were too small to achieve useable results due to poor reproducibility. As a result, another analytical method was developed on a second GC-FID instrument without an MS (Scion 456; Cryo-GC-FID system) that had a pre-concentration trap (15 cm of 1/8" tubing; 1 mL volume) packed with glass beads and cooled with liquid nitrogen that enabled trapping of larger volume samples yielding larger peaks and improved repeatability for all three compounds. The pre-concentration trap was held at -185 °C for 2 mins during sampling prior to being heated to 200 °C and backflushed with carrier gas during the desorption cycle. Separation was achieved using a Rtx-624 capillary column (Restek; 105m × 0.32 mm, df = 1.8 µm). The carrier gas was hydrogen and the flow rate was held constant at 1 mL min$^{-1}$, with a temperature program starting at 30 °C held for 5 minutes, ramped at 25 °C min$^{-1}$ to 200 °C with a final hold of 25 minutes. The total run time was 42 minutes.

Figure 1 shows the chromatograms obtained from both instruments (cryo-GC-FID, blue; GC-MS/FID, red) and demonstrates that all compounds, with the exception of 1,2,4-trimethyl benzene (1,2,4-TMB) and 3-carene, and acetone and dimethyl sulfide (DMS), were baseline separated. The chromatogram in Fig. 1 shows a valley between the 1,2,4-TMB and 3-carene peaks and between the acetone and DMS peaks that provides sufficient separation to obtain robust and repeatable peak areas for all four compounds.

**Figure 1.** Chromatogram of PRM 0819 showing separation of compounds in the GC-FID/MS (red trace) and cryo-GC-FID
(blue trace).

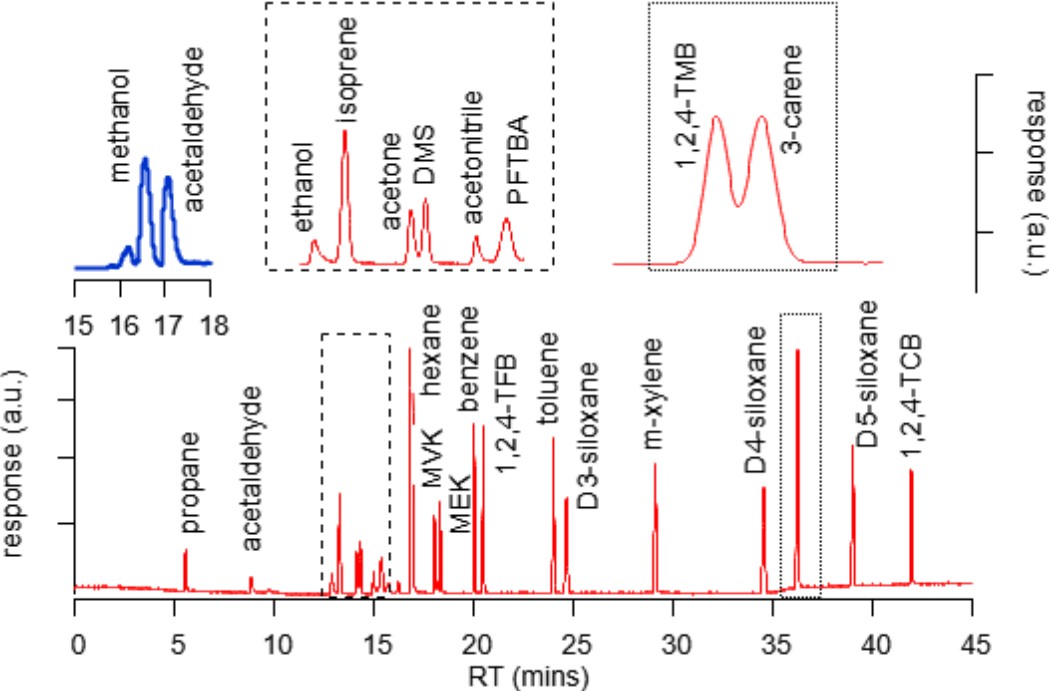


**2.3 Stability assessment**
Stability of all six PRMs were assessed by tracking the ratios of the FID responses of each component relative to an internal
reference that was present in every mixture and which is known to be stable (Rhoderick, 2010; Rhoderick and Lin, 2013;
Worton et al., 2022). Propane was originally included as an internal tracer to monitor stability but as the PTR-MS in $H_3O^+$
mode cannot detect this compound it was replaced by benzene. Benzene is a good internal tracer with stability of > 2 years
that has been well demonstrated relative to propane and hexane for this cylinder type at 5 µmol mol$^{-1}$ with an uncertainty of
0.5 % (Rhoderick et al., 2019). A similar performance would be expected at 1 µmol mol$^{-1}$ and is demonstrated in this work
albeit with an uncertainty of 1 % (Fig. S1, Supporting Information).

All the measurements used for the stability analysis were collected on the same GC-FID/MS instrument with the exception of
methanol and acetaldehyde (Cryo-GC-FID). Both instruments remained unchanged throughout the entire time-period of the
measurements, which spanned more than 4 years. The observed responses for each compound were corrected for differences
in the gravimetric amount fraction and ratioed against the response of the internal reference compound benzene, that was
present in every mixture. The uncertainties in the observed ratios included uncertainties for the gravimetric preparation and
the repeatability of the analyses. The combined standard uncertainties were multiplied by a coverage factor of 2 ($k$=2) providing
a coverage probability of 95 %. The observed ratios were normalised to the average response of all data for that compound to
enable comparisons between compounds with different FID responses. For this analysis the data for all six PRMs were
considered together to enable an understanding of stability across a longer time period than would be possible for a single
PRM. Least squares fit straight-line regressions were modelled to the temporal evolution of the data to determine if there was
any statistically significant change in amount fraction of any of the compounds in the PRMs. The slopes from these regression
analyses were evaluated with an analysis of variance (ANOVA) test using the 'StatsLinearRegression' function in IGOR pro
8.04 (Wavemetrics) (Zar, 1999; Snedecor and Cochran, 1989) to determine whether they were significantly different to zero
using an $F$-test, i.e., no drift in amount fraction during the measurement period (F < Fc).

**2.4 Validation approach**
Five  PRMs (0917a, 0917b, 1218, 0119, 0819, and 0821) were validated against PRM 0819 with the exception of PFTBA and
toluene that were only present in the most recent PRM (0821). PRM 0819 was used as the reference for all the validations
because the parents used for the preparation of this mixture were deliberately different from all other mixtures with the explicit
goal of enabling the most robust validation. All compounds were analysed on the GC-FID/MS system, with the exception of
methanol and acetaldehyde (Cryo-GC-FID). Toluene was validated by comparison against an existing PRM containing BTEX
(benzene, toluene, $m$-xylene, $p$-xylene and $o$-xylene) components that was prepared gravimetrically at NPL in 2018 and had
been independently validated against other PRMs that were internationally compared as part of NPL's participation in key
comparison CCQM-K10.2018 (Cecelski et al., 2022). These BTEX PRMs are known to be stable for more than 5 years and at
the time of the comparisons the BTEX PRM was less than 3 years old.  PFTBA was validated against the gravimetric data
used to make two independent certified reference materials.  The majority of the validation work took place between September
and December 2020 with one in 2019 and 2022, respectively, and three in 2021 (Table S3, Supporting Information). As such
there is an influence of stability on the validation data as the PRMs differed in age at the time of validation.

Each comparison was conducted by running the PRMs (0917a, 0917b, 1218, 0119, and 0821) against PRM 0819 in a repeating
alternating pattern, $(AB)_nA$ where A represents PRM 0819 and B one of the other PRMs ($j$) and with the number of repeats
ranging between 3 and 5 (n = 3 to 5). The ratio in response was determined by dividing B by the average response of the A's
immediately before and after each analysis of B. The average ratio was calculated for each compound based on the number of
repeats along with the associated standard deviation. The assigned analytical value for compound $i$ in PRM $j$ ($x_{u,i,j}$) was
calculated by multiplying the average ratio by the gravimetrical amount fraction ($x_{s,i}$) of compound $i$ in PRM 0819. The relative
difference ($\Delta x$) between the assigned analytical value and the gravimetric value of compound $i$ in PRM $j$ was calculated from:

$$\Delta x \ (\%) = \frac{(x_{u,i,j} - x_{s,i})}{x_{s,i}} \times 100 \qquad\qquad\qquad\qquad\qquad\qquad \text{Eq. 2}$$

The uncertainty in the relative difference combined the standard uncertainty in the repeatability in the analysis with the
gravimetric uncertainty. The combined standard uncertainty was multiplied by a coverage factor of 2 ($k$=2) providing a
coverage probability of 95 %.
**3 Results**
**3.1 Composition**
The PTR-MS transmission curve reference material contains 20 different VOCs that span a wide range of molecular masses,
boiling points, vapour pressures (Table S2, Supporting Information) and functional group classes including alcohol, aldehyde,
ketone, alkene, aromatic, halocarbon and siloxane (Table 2). With the PTR-MS technique, most VOCs are entirely detected at
their protonated mass, as well as a few compounds that fragment during protonation (e.g. monoterpenes, siloxanes, and
isoprene; see Table 1). The compounds were chosen by considering the needs of the PTR-MS user community to cover the
full range of mass-to-charge ratios (m/Q) encountered, their low fragmentation following proton transfer and because many
are of relevance in atmospheric measurements, which was the initial intended target end user group. Other compounds were
included as a consequence of the preparation method, that is the case for *n*-hexane, which is present as the solvent for D3-
siloxane, and propane, which was present in one of the parent mixtures and was originally included as an internal tracer to
monitor stability. The composition evolved over time, as shown in Table 2, with DMS, 1,2,4-TCB, D3-siloxane, toluene and
PFTBA being added at different times, and propane being removed in the final iteration. For D4-siloxane there was a
preparation error, and it was not added to either PRM 1218 or 0119.

An amount fraction of nominally 1 µmol mol$^{-1}$ in a balance gas of nitrogen was selected as a compromise between preparation
complexity and mixture stability. This amount fraction enabled many components to be prepared from parent mixtures of
higher amount fraction ($\geq$10 µmol mol$^{-1}$), which substantially simplifies the preparation process. This amount fraction also
provided a reasonable starting point for stability of the wide range of function groups present in the mixture some of which are
known to have more limited stability at lower abundances fractions (nmol mol$^{-1}$) (Allen et al., 2018).
**3.2 Traceability to the International System of Units (SI)**
Traceability of the primary realisations to the international community through CCQM key comparisons or regional
EURAMET comparisons provides confidence in the accuracy of the amount fractions for all components. SI traceability is
important for underpinning long-term measurements as it provides a stable anchor point with which to reference all
measurements to. Table 1 shows which comparisons underpin the traceability for each of the different components. All the
components are underpinned by at least one CCQM or EURAMET comparison with the exception of 1,2,4-TFB, 1,2,4-TCB
and PFTBA, for which there are currently no existing relevant comparisons.

**3.3 Hierarchies**

Table 2 shows all the parent mixtures and their preparation dates used to prepare all six PRMs (0917a, 0917b, 1218, 0119,
0819, and 0821) and in total 50 different parent mixtures were used.  In general, parent mixtures were similar for PRMs 0917a,
0917b, 1218 and 0119 but were different to PRMs 0819 and 0821 providing independence and thus confidence in the validation
work and in the preparations. There were a few exceptions. For m-xylene the parent used for PRMs 1218 and 0119 was the
same as PRM 0821 but different to 0917a, 0917b and 0821. For 1,2,4-TMB only two parent mixtures were used one for 0917a,
0917b and 0821 and another for 1218, 0119 and 0819. For 3-carene only two parents were used one for 0917a and 0917b and
another for 1218, 0119, 0819 and 0821. For D3-siloxane three parents were used, one for 1218 and 0819, one for 0119 and
another for 0821.

**3.4 PRM Validation**

Figure 2 shows the relative differences ($\Delta x$) determined from Eq. 1 for all compounds using all the validation data obtained
from the 13 comparisons outlined in Table S3 (Supporting Information). In the majority of cases PRM 0819 was used as the
reference to which all others are compared. It was chosen as such because at that time it was the newest PRM to be produced
and was used to benchmark all the others that had already been made. Thus, PRM 0821 was also referenced to PRM 0819 to
provide a link between all six PRMs. All the data shown in Fig. 2 is the FID data from the GC-MS/FID instrument with the
exception of acetonitrile (MS data from the GC-MS/FID instrument), methanol and acetaldehyde (FID data from the cryo-GC-
FID instrument). The MS data is used for acetonitrile because the FID data shows a larger variability, which is likely attributed
to the co-elution of an impurity in the FID that was present at different amount fractions in the different PRMs but we do not
have an conclusive evidence to support this and additional work is needed to confirm. This variability is not observed in the
MS data providing better precision (Fig. S2, Supporting Information).

In general, the data from Fig. 2 could be split into three groups. The first group consisted of propane, isoprene, benzene,
toluene, 3-carene, methanol, acetonitrile, acetaldehyde, *m*-xylene, 1,2,4-TMB, MEK where the spread in the validation data is
within 3 % and these represent components where NPL had substantial prior experience. The second group is acetone, DMS,
MVK and PFTBA where the spread in the validation data is within 5 % and these are relatively new components where
capabilities were developed more recently. Recognising the challenges in preparing PRMs containing siloxanes as a result of
their lower vapour pressures and observing the recent improvements in preparation since 2019, the D4-siloxane and D5-
siloxanes can also be categorised as group 2 after excluding the earliest parent preparations used for 0917a and 0917b in 2017,
which are inconsistent with more recent work as part of the EURAMET 1305 Siloxanes comparison (Van Der Veen et al.,
2022). The final group is comprised of D3-siloxane and 1,2,4-TCB where the spread in validation data is within 10 % and
these compounds represent those which the most challenging to prepare as a result of either unique phase transition properties
or low vapour pressures, respectively. There is an observable bias of about 8 % between two groups of mixtures; one group is
1218 and 0819 and the other is 0119and 0821. This reflects differences between the parent mixtures (2586, 2693 and 3134)
that resulted from the challenges in preparation. Ethanol also sits with this group in part due to the small size of the peak
observed in the GC-MS/FID instrument and because of what looks like an outlier (0119), suggesting some potential losses
during preparation that were unique to this one PRM.

All the FID and supporting MS data for all compounds are shown in Fig. S2 (Supporting Information). No MS data was
available for toluene, 1,2,4-TCB or PFTBA because the relevant single m/Q ions had not been included in the MS single ion
monitoring method at the time of analysis and methanol where the MS signal was too small to provide a reliable response.
Figure S2 shows very good agreement between the FID and MS validation with all components agreeing within the
uncertainties providing confidence in the validation results.

In addition to the observed bias in parent mixtures for D3-siloxane three other parent mixtures were also discovered to be
biased after re-analysis. The observed differences have been corrected for in Fig. 2 and Fig. S2 (Supporting Information). For
methanol, one parent (A410) was confirmed to be 5.0 % high relative to the other parents (A463, A540 and A602) For MVK,
one parent (3070) was confirmed to be 6.3 % low relative to the other parents (2064 and 2088). For 1,2,4-TMB, one parent
D711530 was confirmed to be 6.0 % low relative to D442684 and other in-house standards of 1,2,4-TMB not used in this work
but used to prepare 30 component ozone precursor mixtures at NPL (Grenfell et al., 2010).


**Figure 2.** Relative difference ($\Delta x$) using the FID data (except acetonitrile, which uses the MS data) for all components in five
of the PRMs (0917a, 0917b, 1218, 0119 and 0821) relative to PRM 0819 (solid symbols). The solid black line represents the
average of these validations with the error bar representing the associated expanded uncertainty ($2\sigma$). For D4- and D5-siloxane
the averages do not include the validations from 0917a or 0917b. Methanol and acetaldehyde data are from the cryo-GC-FID
instrument while all others are from the GC-FID/MS instrument. Open symbols represent the original data before correcting
for biases observed in three of the parent mixtures (A410, 5 % low for methanol; 3070, 6.3 % low for MVK and D711530 6
% low for 1,2,4-TMB). PFTBA and toluene were only included in the most recently prepared PRM (0821) and are not present
in 0819. Their validation is described in the text. Supporting validation data from all the MS and FID measurements is shown
in Fig. S2 (Supporting Information).

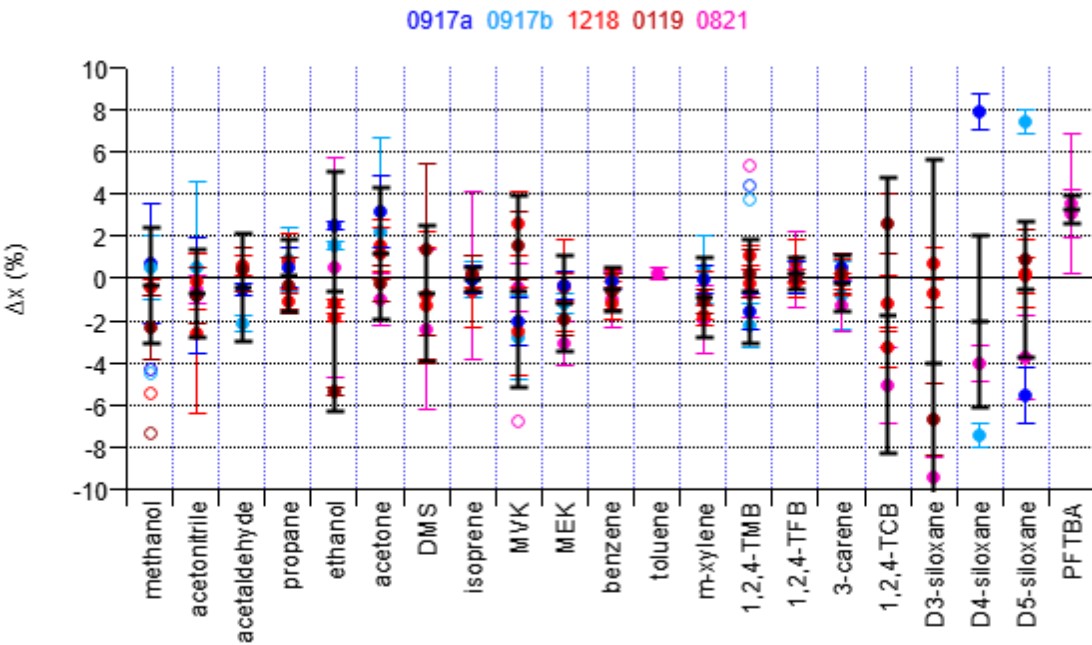



**3.5 CRM Validation**

To enable a more cost effective and timely delivery to end users a certified reference material (CRM) was also developed. In
contrast to the PRMs the CRMs are not prepared by gravimetry but by the direct addition of multicomponent mixtures derived
from the original pure liquids. Further details of the preparation method are given in the Supporting Information
(Supplementary text: Preparation and validation of certified reference materials). The amount fractions for the components in

the CRMs were assigned through analytical comparisons between each CRM and one or more of the PRMs. In this way, preparation is quicker and more cost effective while maintaining the integrity of the values and their traceability. An additional advantage of the CRMs is that because the solid D3-siloxane is dissolved in the other components no *n*-hexane is used which avoids any potential interferences from the presence of reagent ions other than $H_3O^+$ like $O_2^+$ and $NO^+$. Initially with the developed CRM preparation method it was possible to produce mixtures that had blend tolerances of 20 – 30 % (Fig. S3, Supporting Information), which are suitable for end users but work is continuing to improve this with the aim of achieving better than 10 % blend tolerances in the near future. The blend tolerances are just an indication of the repeatability of the preparation process and do not reflect the uncertainties in the assigned value, which are between 3 – 10 % (compound dependent). These uncertainties were dominated by the observed differences between the PRMs.

**3.6 Stability**

Figure 3 shows stability data for four selected compounds; methanol, isoprene, D3-siloxane and PFTBA. These were selected as representative examples of the different observed stability behaviours although the stability data plots corresponding to all compounds are shown in Fig. S1 (Supporting Information). The trendlines from the least squares fit straight-line regressions shown in Fig. 3 and Fig. S1 (Supporting Information) were used to determine the annual drift rates shown in Table 3 and Fig. 4. From the ANOVA test there are statistically significant trends (F > Fc) for 10 of the compounds (methanol, acetonitrile, acetaldehyde, ethanol, acetone, DMS, isoprene, MVK, benzene, D3-siloxane) but these trends are small (< 2 % $yr^{-1}$) except for methanol, acetonitrile and PFTBA.

Methanol and acetaldehyde were the only two components that were measured on the cryo-GC-FID and hence the datasets are more limited. A result is that there is no overlap between the three PRMs so any systematic differences between them may result in an artificial bias, which may exacerbate any stability trend. More work is needed to confirm this. The stability data for D3-siloxane reflects the observed validation bias and shows two clear trends; one for 1218 and 0819 and the other for 0119 and 0821. For the regression analysis and drift calculations these have been treated independently (Table 3).

All compounds, with the exceptions of methanol, acetonitrile and PFTBA, show trends similar to isoprene with good stability and annual drift rates of < 3 % $yr^{-1}$ (Table 3). For acetonitrile the large spread in validation data (FID data; Fig. S2, Supporting Information) leads to a noisy stability dataset that may play a role in the larger observed drift rate or this component maybe less stable. As PFTBA was only included in the last PRM (0821) the stability data only represents about half a year and extrapolating the current trend to 1 and 2 years results in a drift rate that is not accurate as interpolation of the data would suggest no statistical change in amount fraction and minimal drift. More data is needed to confirm the longer-term stability behaviour of PFTBA.

**Figure 3.** Stability of normalised response with time for four selected compounds relative to benzene, methanol (top left),
isoprene (top right), D3-siloxane (bottom left) and PFTBA (bottom right) for all six PRMs (solid symbols). The open symbols
(methanol; top left) show the original data before being corrected for an observed 5.0 % bias in the parent mixture (A410).
The best fit curves from least squares straight line regression analyses are shown (solid black line) along with the 95 %
confidence interval of the fits (shaded area). The slope, intercept and F-statistic data from the regression analyses are shown
in Table 3. Stability plots for all compounds are shown in Fig. S1 (Supporting Information).

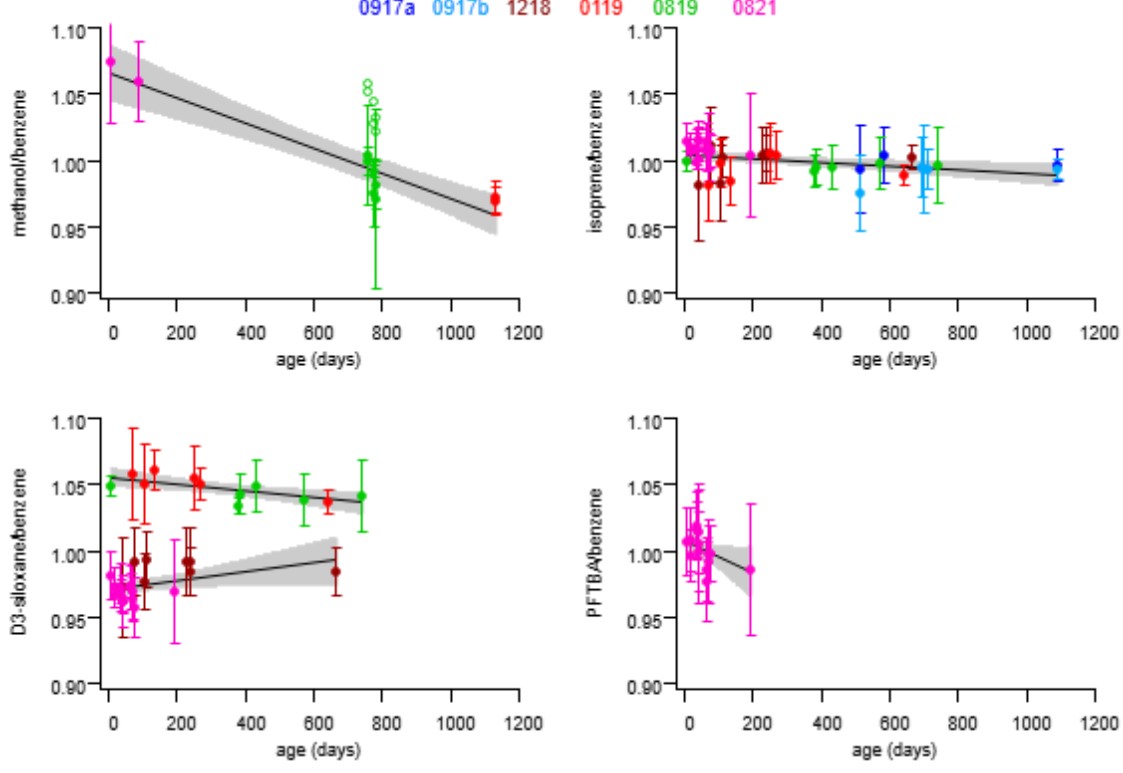



Given the age differences between the different PRMs at the time of validation (233 – 709 days; Table S2, Supporting
Information) it is not possible to deconvolute the contributions of stability and preparation to the observed validation
differences. However, Fig. 4 shows that for the majority of compounds there is good agreement between the observed
average validation data and the calculated drift for over 1 – 2 years, with the exception of methanol, acetonitrile and PFTBA,
which differ for the reasons discussed previously. These observations are consistent with the age differences of the different
PRMs at the time of validation indicating that stability was likely the major driver between the observed validation
differences.
**Table 3.** Summary of the results of the least squares straight-line regression analysis for all stability data shown in Fig. 3 and
Fig. S2 (Supporting Information). Results are shown for the slope ($\pm 2\sigma$), intercept ($\pm 2\sigma$), ANOVA test statistics (F and Fc)
used to evaluate the presence of a statistically significant trend (F > Fc), the calculated annual drift ($\pm 2\sigma$) determined from the
linear fit and the average of the validation data ($\pm 2\sigma$), also shown in Fig. 2.

| compound | slope ($\times 10^{-5}$) | intercept | F | Fc | Drift (%/yr) | Avg valid. (%) |
|---|---|---|---|---|---|---|
| methanol[a] | -9.539 ± 2.700 | 1.067 ± 0.021 | 57.005 | 5.318 | -3.48 ± 0.42 | -0.37 ± 2.77 |
| acetonitrile | -12.328 ± 6.128 | 1.036 ± 0.026 | 16.530 | 4.085 | -4.50 ± 0.94 | -0.72 ± 2.08 |
| acetaldehyde[a] | -5.345 ± 2.800 | 1.037 ± 0.022 | 19.699 | 5.318 | -1.95 ± 0.44 | -0.40 ± 2.53 |
| propane | 0.653 ± 5.393 | 0.997 ± 0.027 | 0.062 | 4.225 | 0.24 ± 1.97 | 0.16 ± 1.66 |
| ethanol | -7.841 ± 7.55 | 1.023 ± 0.032 | 4.405 | 4.085 | -2.86 ± 0.36 | -0.61 ± 5.64 |
| acetone | 3.462 ± 3.206 | 0.990 ± 0.013 | 4.765 | 4.085 | 1.26 ± 0.86 | 1.18 ± 3.08 |
| DMS | 2.441 ± 2.351 | 0.995 ± 0.007 | 4.473 | 4.149 | 0.89 ± 2.24 | -0.76 ± 3.22 |
| isoprene | -1.338 ± 0.975 | 1.004 ± 0.004 | 7.690 | 4.085 | -0.49 ± 1.17 | -0.04 ± 0.60 |
| MVK | -3.523 ± 2.564 | 1.010 ± 0.011 | 7.708 | 4.085 | -1.29 ± 0.94 | -0.61 ± 4.50 |
| MEK | 0.575 ± 1.967 | 0.998 ± 0.008 | 0.349 | 4.085 | 0.21 ± 0.36 | -1.23 ± 2.25 |
| Benzene[b] | 1.329 ± 0.983 | 0.996 ± 0.004 | 7.456 | 4.085 | 0.49 ± 0.18 | -0.48 ± 1.07 |
| Toluene[c] | -3.546 ± 4.536 | 1.002 ± 0.004 | 2.902 | 4.747 | -1.30 ± 1.66 | 0.19 ± 0.29 |
| m-xylene | 0.129 ± 2.034 | 1.000 ± 0.009 | 0.016 | 4.085 | 0.05 ± 0.74 | -0.87 ± 1.88 |
| 1,2,4-TMB | -0.870 ± 5.155 | 1.003 ± 0.022 | 0.116 | 4.085 | -0.32 ± 1.69 | -0.57 ± 2.42 |
| 1,2,4-TFB | -1.373 ± 1.448 | 1.004 ± 0.006 | 3.672 | 4.085 | -0.50 ± 2.05 | 0.27 ± 0.74 |
| +3-carene | -0.734 ± 4.631 | 1.002 ± 0.019 | 0.103 | 4.085 | -0.27 ± 2.84 | -0.25 ± 1.33 |
| 1,2,4-TCB | 4.512 ± 6.455 | 0.991 ± 0.018 | 2.027 | 4.149 | 1.65 ± 1.16 | -1.73 ± 6.56 |
| D3-siloxane[d] | -2.641 ± 1.740 | 1.056 ± 0.007 | 11.444 | 4.965 | -0.96 ± 0.29 | -4.02 ± 9.67 |
| | 3.195 ± 3.220 | 0.970 ± 0.007 | 4.287 | 4.351 | 1.17 ± 0.56 | |
| D4-siloxane[e] | 4.799 ± 4.300 | 0.988 ± 0.012 | 0.765 | 4.225 | 1.75 ± 0.74 | -2.03 ± 4.06 |
| D5-siloxane[e] | 2.066 ± 0.390 | 0.985 ± 0.026 | 1.833 | 4.085 | 0.75 ± 1.68 | -0.49 ± 3.27 |
| PFTBA[c] | -12.045 ± 13.440 | 1.007 ± 0.010 | 3.813 | 4.747 | -4.40 ± 1.66 | 3.31 ± 0.70 |

[a]The GC-FID data for methanol and acetaldehyde was too small to be quantified so this data is from the cryo-GC-FID data and is limited. [b]Benzene stability
was determined relative to isoprene. [c]Toluene and PFTBA were only included in the most recent PRM so the assessment of stability is limited in its duration
to only 200 days. [d]There was a clear bias between several of the PRMs caused by differences in the parent mixtures used so the trends were fitted to the two
obvious groupings. [e]Data from 0917a and 0917b were excluded from the regression analysis.

**Figure 4.** Comparison of 1-year (filled grey squares) and 2-year (open grey squares) drift rates, calculated from the data in
Table 3, with the average validation data (black bars) taken from Fig. 2. For D3-siloxane there are two datapoints for the drift
correspond to the two regressions shown in Table 3. The error bars represent the associated expanded uncertainties,
representing the 95 % confidence limit.

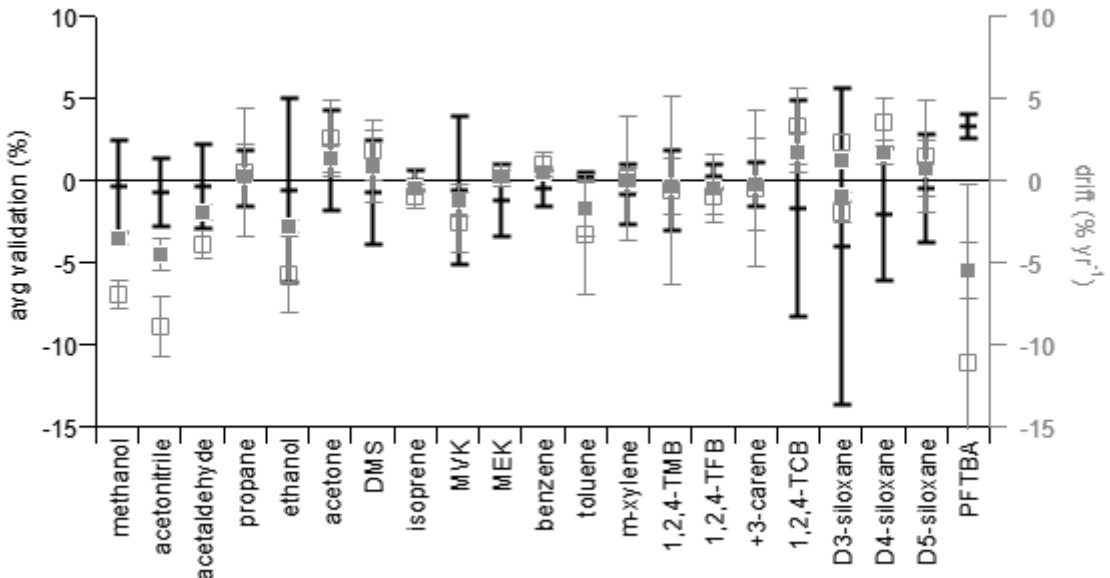


**4 Conclusions**
In this work the development of new primary reference materials (PRMs) and certified reference materials (CRMs) for
constraining the mass dependent transmission curve of PTR-MS instruments have been described along with an evaluation of
the validation and stability of the PRMs and the repeatability in preparation (blend tolerances) for the CRMs. Six of these
PRMs have been prepared to date from a suite of 50 parent mixtures and these have been used to value assign more than 10
CRMs that have been disseminated to end users. In general, there is evidence of very good agreement for the majority of
components that supports the robustness of the preparation and 2 years of stability. Challenges were observed in preparation
for the least volatile compounds especially for D3-siloxane due to it being a solid at room temperature and pressure. More
work is needed to better describe the long-term stability of methanol, acetonitrile and PFTBA. This work highlighted several
challenges in analysis that could be resolved by the development of a new analytical method utilised a single instrument
equipped with both a preconcentration trap and dual detector setup (MS and FID). This work demonstrates what is currently
possible with respect to composition, amount fraction, uncertainty and stability and provides an important reference to which
other gas standards that are in use with the PTR-MS can be compared and benchmarked to verify their accuracy to further
improve the comparability of PTR-MS measurement data.

In the short term (next 5 years) the implementation of an SI traceable transmission curve reference material, such as the one
described in this work, using a method similar to that described in Holzinger et al., 2019 is the most pragmatic approach to
directly address improving the accuracy of quantitation and comparability between different PTR-MS instruments and users.
This reflects the challenges and complications of rapidly developing a universally accepted calibration system based on pure
liquids that is SI traceable. The use of a SI traceable reference material to properly constrain the transmission curve provides
a readily applicable framework to ensure confidence in temporal and spatial data to support the use of PTR-MS in a broad
range of application areas. The use of the transmission curve reference material approach should be seen as a pre-requisite and
a complement to additional future efforts to provide alternative calibration efforts for specific target compounds where
uncertainties of better than 30 % are needed. Alternative approaches would certainly be necessary for those compounds that
are unsuitable for inclusion in high pressure gas standards possibly as a result of very low vapour pressures or other
complicating factors such as chemical compatibility with other compounds.

Future work to improve the uncertainty of individual components that have the greatest influence on the transmission curve fit
would have the biggest influence on the accuracy and repeatability of the transmission curve retrieval thereby maximising the
impact of future improvements for the PTR-MS user community. For PTR-MS instruments that utilise time of flight mass
spectrometers the focus would be on improving the uncertainty of the largest molecular weight components specifically the
D3-, D4-, D5-siloxanes and 1,2,4-TCB, which represent the greatest challenges in preparation due to their low vapour
pressures.
**Data availability**
All data used to produce the figures in this paper are available on request.
**Competing interests**
The authors declare that they have no conflict of interest.

## Author contributions

DRW and RH conceptualized the work. DRW processed the data, produced the figures and tables and wrote the paper. RH provided inputs to define the composition of the PTR-MS reference material and contributed to the writing of the paper. SM developed novel methods for the preparation of primary and certified reference materials, planned and prepared all reference materials, conducted all the validation and stability data collection and contributed to the writing of the paper. KOD contributed to the preparation of reference materials, reprocessed some of the stability data and worked with SM to prepare the certified reference materials. All authors reviewed the paper.

## Financial support

NPL acknowledges funding from the Department of Business Energy and Industrial Strategy (BEIS) National Measurement System. RH acknowledges funding from the EMPIR programme (19ENV06 MetClimVOC), co-financed by the Participating States and from the European Union's Horizon 2020 research and innovation programme, and from the European Union's Horizon 2020 research and innovation programme (ACTRIS-2) under grant agreement no. 654109.

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
