# Peer review of "Development of an SI-traceable transmission curve reference material"

_Atmospheric Measurement Techniques, 2022_

## Referee Comment (RC1)

**Referee comments to**

**„Development of an SI-traceable transmission curve reference material to improve comparability of proton transfer reaction mass spectrometry measurements" by David Robert Worton, Sergi Moreno, Kieran O'Daly and Rupert Holzinger**

**by Wiebke Scholz**

The manuscript by Worton et al. describes the production and evaluation of several multi-component gaseous primary reference materials (PRM) with the aim to allow PTR-MS users to better constrain the transmission curves of their respective instruments. The manuscript discusses the challenges of including low-volatility compounds, but manages well to reproduce their concentrations in the different PRMs with uncertainties typically below 10%.
The quantitative addition of such high-mass low-volatility molecules to gas-standards will be of great use for the PTR-MS user community in general and is also important for atmospheric measurements. Challenges in the quantification of several compounds were overcome by using a combination of GC-MS and GC-FID and in some cases Cryo-GC-FID.
The PRMs described were prepared following standardized procedures and evaluation results are presented in great detail and with particular dedication to precision and uncertainty analysis in a well-structured manner. However, the manuscript was in parts difficult to understand on the first read, because abbreviations were used excessively and partly confusing.
The paper gives insight into the reproducability and stability of gas standards that will be very valuable to end users. Transmission-curve contrainments with one single reference material can simplify the lifes of many PTR-MS users around the world and enhance data comparability and quantification.

I therefore suggest that the manuscript is published in AMT after some minor comments have been addressed:

1. The GC-MS method used is not fully described. Please specify especially the type of ionization (electron impact, chemical ionization …) (lines 94 ff.)

2. How was the separation of the 3-carene and 1,2,4-TMB peak treated? Was a multipeakfit performed and the data corrected accordingly or do the two compounds influence the other's signal? (lines 96 ff.)
In line 144, the baseline separation issue is mentioned for Acetone and DMS as well, but it is missing in lines 96 ff.

3. For the Cryo-GC-FID, the volume of the loop, and the trapping / heating cycle are not described (l. 105 ff.)

4. Why are the FID data on acetonitrile so much more noisy than the acetonitrile GC-MS data and also so much more noisy than the methanol FID data, although the latter should give a smaller FID signal? Consider adding a sentence or two on this matter.

5. Please make sure, the abbreviations are used consistently, as these were confusing in parts:
   - Line 57: NPL is not defined
         (don't let the reader search for the hint in Dave Worton's email address…)
   - Line 154: Is PTRMS NPL PRMs a subgroup of NPL PRMs?
         If yes, please define, if no, please remove the „PTRMS"
   - Line 152: Mentioning of the certified primary reference material without the short-form
         „CRM" used in the supplement.
   - Also, the supplement about the certified reference material with fig. S3 is not
   mentioned in the main text and so the difference between certified reference material
   and PRM remains unclear in the main text. It appears, as if the uncertainty for CRMs
   is larger than for PRMs due to the simpler preparation method. What does that mean
   for the PFTBA validation?
   - In this context, there is also missing information on the BTEX NPL PRM in the main
   text (especially its age, and if it is produced also gravimetrically)
   - Text and Fig. S3 in the supplement:
   Decide on CRM, PTRMS CRM or NPL CRM to support the reader, except you want
   to address differences, that then should be clearly defined.
   - The names of the different reference materials are not clearly motivated and
   therefore reduce the readability.
   I suggest changing them so they e.g. contain the month and year of production. To
   guide the rader for a faster understanding of Table 2 consider coloring the Cylinder
   IDs in table 2 as you do in the figures.

6. line 158 ff:
   - did you really measure „A" always twice directly after each other or were the repeats
   performed on different dates / which longer breaks in between? If so, add this. Otherwise,
   consider correcting to $(AB)_nA$
   - What is meant by „bracketing" between the nearest neighbors? Consider using a more
   common word that is clear also for non-english natives.
   - Add to Eq. 1 that it is calculating the relative difference in %

7. l. 198 „A638" is mentioned twice

8. The footnotes in table 3 (a,b,c,...) do not correspond to the markers within table 3 (1,2,3,...)

9. it is not clear to me, why the reference would be a PRM that is produced from many old parent
mixtures. Thus I wonder, what is the advantage of 2819 compared to using D961492? (line 149)
In general, only little information has been given on the stability of the parent mixtures.

10. In fig S2, D4-siloxane and D5-siloxane, biased data do not appear corrected again (as they do
e.g. in the plot for 1,2,4-TMB). Please add a short comment regarding this to the figure caption.

11. The PTR-MS transmission curve constrainments would be performed with the CRMs appearing only in the supplement fig. S3 in the future, as I understood. The preparation repeatability is reduced to 20-30%. Is this still sufficient to determine a transmission curve, which usually varies within a factor 1.5-2 or will the CRMs be cross-evaluated against the PRMs? Please clarify.

12. Another technique to determine the transmission curve is the addition of single species to the sample air in step-wise increasing concentrations and observe the change in primary ion signal and the ion signal of the sample molecule and repeating this for multiple compounds with different masses. This technique is time-consuming for the end-user but gives generally good results. By having one gas standard that combines many compounds spanning the whole mass range, the technique of comparing with the primary ion signal is certainly not possible anymore, which requires the CRM to really be precise and contain species that are detected at the kinetic limit. I am looking forward to your discussion of this point.

A short „how-to" for the usage of your CRMs for transmission curve correction, potential error sources and prerequirements would add great value to your manuscript, especially for new users.

---

## Author Comment (AC1)

**Response to reviewer 1**

The authors thank the reviewer for the positive response and thorough comments that have improved the manuscript. The point-by-point response is below. The reviewers' comments are numbered and are in black font, the authors responses are also numbered and in blue font to ease readability.
* * *
**Comments from Reviewer**

The manuscript by Worton et al. describes the production and evaluation of several multicomponent gaseous primary reference materials (PRM) with the aim to allow PTR-MS users to better constrain the transmission curves of their respective instruments. The manuscript discusses the challenges of including low-volatility compounds, but manages well to reproduce their concentrations in the different PRMs with uncertainties typically below 10%. The quantitative addition of such high-mass low-volatility molecules to gas-standards will be of great use for the PTR-MS user community in general and is also important for atmospheric measurements. Challenges in the quantification of several compounds were overcome by using a combination of GC-MS and GC-FID and in some cases Cryo-GC-FID. The PRMs described were prepared following standardized procedures and evaluation results are presented in great detail and with particular dedication to precision and uncertainty analysis in a well-structured manner. However, the manuscript was in parts difficult to understand on the first read, because abbreviations were used excessively and partly confusing. The paper gives insight into the reproducibility and stability of gas standards that will be very valuable to end users. Transmission-curve constrainments with one single reference material can simplify the lives of many PTR-MS users around the world and enhance data comparability and quantification. I therefore suggest that the manuscript is published in AMT after some minor comments have been addressed.

**R1.1** The GC-MS method used is not fully described. Please specify especially the type of ionization (electron impact, chemical ionization …) (lines 94 ff.)

Added the following text 'electron ionisation (70 eV)' before mass spectrometer on line 94.

**R1.2** How was the separation of the 3-carene and 1,2,4-TMB peak treated? Was a multipeakfit performed and the data corrected accordingly or do the two compounds influence the other's signal? (lines 96 ff.) In line 144, the baseline separation issue is mentioned for Acetone and DMS as well, but it is missing in lines 96 ff.

While the 3-carene and 1,2,4-TMB peaks and the acetone and DMS peaks are not baseline resolved the separations were observed to be very consistent throughout all the chromatograms collected and this is described already in the manuscript between lines 114 and 118. A multipeakfit deconvolution was not used.

**R1.3** For the Cryo-GC-FID, the volume of the loop, and the trapping / heating cycle are not described (l. 105 ff.)

These details have now been added to the text.

**R1.4** Why are the FID data on acetonitrile so much more noisy than the acetonitrile GC-MS data and also so much more noisy than the methanol FID data, although the latter should give a smaller FID signal? Consider adding a sentence or two on this matter.

It was not clear as to what the main reason for the noisier GC-FID data. It is possible that a co-eluting impurity that was present at different amount fractions in the different PRMs could be an explanation but we do not have an conclusive evidence to support this suggestion. We have added some text eluding to this in the main manuscript.

**R1.5** Consistent use of abbreviations:
Line 57: NPL is not defined (don't let the reader search for the hint in Dave Worton's email address…)

Line 154: Is PTRMS NPL PRMs a subgroup of NPL PRMs? If yes, please define, if no, please remove the "PTRMS"

Line 152: Mentioning of the certified primary reference material without the short-form "CRM" used in the supplement.

Apologies for the confusion here. We have rechecked the manuscript and have refined the abbreviations to be consistent throughout and to be simpler so as to convey only the most pertinent information (i.e., PRM, CRM).

**R1.6** Also, the supplement about the certified reference material with fig. S3 is not mentioned in the main text and so the difference between certified reference material and PRM remains unclear in the main text. It appears, as if the uncertainty for CRMs is larger than for PRMs due to the simpler preparation method. What does that mean for the PFTBA validation?

To clarify this we have added the anew section 3.5 CRM Validation and have added the following text to the main manuscript citing the discussion of the CRM in the supplement as well as explaining the difference between a CRM and PRM: "**To enable a more cost effective and timely delivery of the PTR-MS transfer standard an NPL certified reference material (CRM) was also developed. In contrast to the NPL PRMs the CRMs are not prepared by gravimetry and are produced by the direct addition of multicomponent mixtures derived from the original pure liquids. Further details of the preparation method are given in the Supporting Information (Supplementary text: Preparation and validation of certified reference materials). The amount fractions for the components in the CRMs were assigned through analytical comparisons between each CRM and an NPL PRM. In this way, preparation is quicker and more cost effective while maintaining the integrity of the values and their traceability. Initially with the developed CRM preparation method it was possible to produce mixtures that had blend tolerances of 20 – 30 % (see Figure S3, Supporting Information), which are suitable for end users but work is continuing to improve this with the aim of achieving better than 10 % blend tolerances in the near future. The blend tolerances are just an indication of the repeatability of the preparation process, the amount fractions of each CRM are analytical assigned by comparison to one or more NPL PRMs. The uncertainties in the amount fractions for the CRMs are independent on the blend tolerances and where between 2 – 10 %, compound dependent, and these uncertainties were predominantly drive by the observed differences from the preparation of the PRMs.** "

The CRMs have a larger uncertainty than the PRMs because in addition to the uncertainties in the gravimetric preparation data and the validation the uncertainty from the analytical value assignment of the CRMs from the PRMS also needs to be included. However, for the PFTBA, the validations were done using the gravimetric data of the preparations. As such this was developed using a primary method (the gravimetric method). This was done by addition of PFTBA by two separate weighed loop additions to two independent cylinders, which were then validated through analytical

comparisons to the values expected from the gravimetric data as such it was validated in line with the PRMs.

**R1.7** In this context, there is also missing information on the BTEX NPL PRM in the main text (especially its age, and if it is produced also gravimetrically)

The following information on the BTEX PRM has been added to the main text at line XXX: "Toluene was validated by comparison against an existing NPL PRM containing BTEX (benzene, toluene, *m*-xylene, *p*-xylene and *o*-xylene) components **that was prepared gravimetrically in December 2018 and had been independently validated against other NPL PRMs that were internationally compared as part of key comparison CCQM-K10.2018 (Cecelski et al., 2022). These BTEX PRMs are known to be stable for more than 5 years and at the time of the comparisons the BTEX PRM was less than 3 years old.**"

**R1.8** Text and Fig. S3 in the supplement: Decide on CRM, PTRMS CRM or NPL CRM to support the reader, except you want to address differences, that then should be clearly defined.

Apologies for the confusion. These have all been changed to NPL CRM throughout.

**R1.9** The names of the different reference materials are not clearly motivated and therefore reduce the readability. I suggest changing them so they e.g. contain the month and year of production. To guide the reader for a faster understanding of Table 2 consider colouring the Cylinder IDs in table 2 as you do in the figures.

The names of the different reference materials came originally from the unique identifiers of the actual cylinders used but we acknowledge that these are not meaningful to the reader so we have changed them using the month and year suggested by the reviewer. Thus A574, A578, A638, A643, 2819, D961492 become 0917a, 0917b, 1218, 0119, 0819, 0821, the colour coding suggestion has also been adopted in Table 2.

**R1.10** line 158 ff: - did you really measure "A" always twice directly after each other or were the repeats performed on different dates / which longer breaks in between? If so, add this. Otherwise, consider correcting to (AB)nA.

What is meant by „bracketing" between the nearest neighbors? Consider using a more common word that is clear also for non-english natives.

Add to Eq. 1 that it is calculating the relative difference in %

We have corrected the repeating pattern to (AB)nA. We have removed the reference to nearest neighbours and added the following text to clarify what is meant here: "The ratio in response was determined by **dividing B by the average response of the A's immediately before and after each analysis of B.**", We have added the % sign into Eq. 1.

**R1.11** l. 198 „A638" is mentioned twice

This was a typo and should read A643 (now 0119 in the new naming convention, see R1.9 above).

**R1.12** The footnotes in table 3 (a,b,c,...) do not correspond to the markers within table 3 (1,2,3,...)

This has been corrected.

**R1.13** it is not clear to me, why the reference would be a PRM that is produced from many old parent mixtures. Thus I wonder, what is the advantage of 2819 compared to using D961492? (line 149) In general, only little information has been given on the stability of the parent mixtures.

The reason that 2819 is used as the reference is because many of the validations between the PRMs took place before D961492 had been produced so comparing to 2819 provides a consistent link between all of the PRM mixtures. The parents have been produced and validated over many years and form part of the National Standards, maintained by NPL. While some have been produced a number of years ago they have been measured on numerous occasions against newly prepared mixtures that were destined for third parties as part of our measurement service offerings. These validations and those presented in this work demonstrate that the parents are stable at least on the timeframe of the PRM produced in this work and probably much longer. Typically, we observed stabilities that are longer for higher amount fraction mixtures compared to those at lower amount fractions. The following text has been added to section 3.4 (PRM validation) to clarify the choice of 2819 as the reference PRM: "**In the majority of cases PRM 0819 is used as the reference to which all others are compared. It was chosen as such because at the time it was the newest PRM to be produced and was used to benchmark all the others that had already been made. Thus, PRM 0821 was also referenced to PRM 0819 to provide a link between all six PRMs**."

**R1.14** In fig S2, D4-siloxane and D5-siloxane, biased data do not appear corrected again (as they do e.g. in the plot for 1,2,4-TMB). Please add a short comment regarding this to the figure caption.

Noted. Additional text has been added in the caption and is shown in bold here: "The open symbols show the original data before being corrected for biases in the parent mixtures (for methanol, MVK and 1,2,4-TMB) **or which has been excluded from the regression analysis (for D4-siloxane and D5-siloxane)** as discussed in the text."

**R1.15** The PTR-MS transmission curve constrainments would be performed with the CRMs appearing only in the supplement fig. S3 in the future, as I understood. The preparation repeatability is reduced to 20-30%. Is this still sufficient to determine a transmission curve, which usually varies within a factor 1.5-2 or will the CRMs be cross-evaluated against the PRMs? Please clarify.

This is correct. The intention is to disseminate the CRMs to end users and for the PRMs to be maintained for the purposes of value assignment of the CRMS. The preparation repeatability or blend tolerance of the CRMs is just an indication of the repeatability of the process. The amount fractions of the CRMs are based on analytical values determined by comparison to an NPL PRM. See response to R1.5 above.

**R1.16** Another technique to determine the transmission curve is the addition of single species to the sample air in step-wise increasing concentrations and observe the change in primary ion signal and the ion signal of the sample molecule and repeating this for multiple compounds with different masses. This technique is time-consuming for the end-user but gives generally good results. By having one gas standard that combines many compounds spanning the whole mass range, the technique of comparing with the primary ion signal is certainly not possible anymore, which requires the CRM to really be precise and contain species that are detected at the kinetic limit. I am looking forward to your discussion of this point. A short "how-to" for the usage of your CRMs for transmission curve correction, potential error sources and pre-requirements would add great value to your manuscript, especially for new users.

Besides being time consuming, the compound-by-compound method has the disadvantage that the high product ion concentration interfere with the transmission of the primary ions. Especially in

new-generation instruments counter intuitive behaviour has been observed (e.g., increasing primary ion signal as the compound concentration is increased). So, while this method produces acceptable results in older instruments, this is not an option for instruments featuring more complex ion optics. For the 'how-to' question we refer to Holzinger et al. (2019) where the use of the standard is discussed in detail. We have added the following text to the manuscript at the end of the introduction to direct the reader to Holzinger et al., 2019 for details on how to use the CRMs for transmission curve correction: "**For details on how to use the RMs to constrain the PTR-MS transmission curve the reader is directed to H** "

---

## Author Comment (AC2)

**Response to reviewer 2**

The authors thank the reviewer for the positive response and thorough comments that have improved the manuscript. The point-by-point response is below. The reviewers' comments are numbered and are in black font, the authors responses are also numbered and in blue font to ease readability.
* * *
**Comments from Reviewer**

Worton et al. present a neat development of a state-of-the-art multicomponent gas standard and evaluates its accuracy and stability. The major application of the standard will find in constraining the transmission of PTR-MS instruments which in turn will help in more accurate quantification of uncalibrated compounds based on the proton transfer reaction theory. The standard looks extraordinary in its meticulousness of preparation, is SI-traceable and well characterized in terms of stability of the included compounds. An impressive achievement was to embrace complex chemical compositions varying many orders of magnitude in vapor pressures providing unprecedented mass range of 32 to 671 Da. While the manuscript is generally well written and will be useful probably beyond the PTR-MS community, it has a potential for further enhancements of its clarity. I made just a few relatively minor comments which hopefully can be addressed in the revised version.

**General**

**R2.1** It is somewhat surprising that the paper assumes pre-existing knowledge from a general AMT reader about concepts such as mass spectrometer's transmission. I think it would be helpful for the novice PTR-MS audience as well as general community a paragraph or a section that explains the basics of transmission and then refer the reader for more details to Holzinger et al. (2019). Additionally, concepts used inconsistently (e.g. transmission curve and transfer curve) may unnecessarily increase readers' mental processing time.

Noted about inconsistent use of concepts. Transfer has been changed to transmission curve throughout the manuscript. The following text has been added to the manuscript in the introduction at line 55 to explain the basics of the transmission: "**The basis for this is that the amount fraction of compound R ([R]) can be determined from (Taipale et al., 2008):**

$$[R] = \frac{1}{k\Delta t} \times \frac{I(RH^+)}{T(RH^+)} \times \left(\frac{I(H_3O^+)}{T(H_3O^+)}\right)^{-1} \qquad (1)$$

**Where k is the proton transfer reaction rate coefficient, Δt is the reaction time, I(RH⁺) and I(H₃O⁺) are the observed ion count rates for the protonated ion of compound R (RH⁺) and the hydronium ion (H₃O⁺), respectively. T(RH⁺) and T(H₃O⁺) are the transmission efficiencies for RH⁺ and H₃O⁺ ions, respectively. The transmission coefficients are predominantly mass dependent, but they can also vary in time (De Gouw et al., 2003; Ammann et al., 2004; Steinbacher et al., 2004). Proton transfer reaction rate coefficients can be measured and/or reasonably well predicted using quantum methods (Zhao and Zhang, 2004). If specific rate coefficients are agreed within the community for specific compounds and are widely used this would negate the role of different rate constants on measurement comparability (Table S1, Supporting Information). The reaction time and observed ion count rates are all measured parameters leaving just the transmission coefficients as variables required for quantitative measurements without specific calibrations. Cappellin et al. (2012) demonstrated the quantitative properties of one type of PTR-MS instrument by assuming a theoretical transmission based on the duty cycle of the time-of-flight mass analyser. However, for newer generation instruments that employ advanced ion optics to improve sensitivity, it is**

necessary to determine the mass-dependent transmission experimentally as the transmission of the system diverges from theory at low masses. Deviations can also occur at high masses due to poor tuning and/or ageing of the ion detection system (Müller et al., 2014).

There are several highly cited publications that explore best practices in PTR-MS measurements (e.g., Blake et al., 2009; De Gouw and Warneke, 2007; Yuan et al., 2017), including methods to calibrate and retrieve the mass dependent transmission (Taipale et al., 2008). However, many of these methods are slow and labour intensive and as a result calibrations and transmission curve retrievals are not performed frequently enough. This has limited the application of PTR-MS to mostly short campaign-scale intensive deployments and only a few groups have utilised PTR-MS for long-term studies (Holzinger et al., 2006; Taipale et al., 2008). However, recent work by Holzinger et al., (2019) has demonstrated: (i) a new method to retrieve the mass-dependent transmission from fast calibrations that should enable more frequent calibrations and (ii) the validity of a simple reaction kinetics approach to quantify measurements of uncalibrated compounds from different PTR-MS instruments with an accuracy of ≤ 30 % provided the transmission curve is accurately constrained.”

**R2.2** The inclusion of volatile cyclic siloxanes to gas standards is phenomenal but is not new and was already neatly conducted by other vendors with high reputation in the VOC community such as Apel-Riemer Environmental, Inc. who have been at the forefront of preparing those mixtures at a 5% accuracy confirmed by the GC measurement in about hundreds of PTR-MS papers (e.g. Tang et al., 2014; Werner et al., 2021). It is unclear how the NPL standard stands out because it does not compare other standards used by the community which seems like a lost opportunity for this otherwise excellent paper.

What is unique here is the SI traceable nature of the majority of components. To be able to demonstrate SI traceability through international key comparisons with other NMIs is key here. This ensures the accuracy of the components, which is of critical importance for both comparability and also for ensuring the accuracy (trueness) of the reference material remains constant with time. As NPL is not part of the PTR-MS community it is difficult for us to make comparisons to standards that are actively used in this community and we rely on end users to make those comparisons and report those results. We would certainly encourage the community to make those comparisons or to engage with us to provide us with their standards to make such comparisons.

**R2.3** Although the paper shows exemplary progress for future transmission measurements, I think it could be made clear that the NPL standard is not trying to monopolize the gas standard transmission market. As the fair comparison with other standards has not been provided, a note mentioning that other standards can also be potentially appropriate and useful for transmission measurements would be reassuring.

Indeed this is not the intention. In the conclusion we have added some text that eludes to the use of other standards. The added text is: “**This work demonstrates what is currently possible with respect to composition, amount fraction, uncertainty and stability and provides an important reference to which other gas standards that are in use with the PTR-MS can be compared and benchmarked to verify their accuracy to further improve the comparability of PTR-MS measurement data.**”

**R2.4** The reference list seems somewhat modest and generally not acknowledging the progress in overcoming challenges in transmission measurements and calibrations which have been widely used by the PTR-MS community for almost 3 decades. I encourage making a stronger connection to the

PTR-MS classic (e.g. Taipale et al., 2008) and recent literature (not only Holzinger et al., 2019) and further emphasizing the novelty and advancements that the new standard might offer.

We have added additional references and text in the introduction after line 55 to address this, see the response to R2.1 above which contains the added text.

**Specific**

**R2.5** I agree that n-hexane was a relatively good choice for the D3 solvent, but the statement in the SI is misleading about n-hexane undetectability by PTR-MS: "because the proton affinity of n-hexane is less than water and therefore does not undergo proton transfer and is not detectable by PTR-MS when operating in the H3O+ mode.". What I want to remind is that there is no 100% pure H3O+ mode so all PTR-MS instruments operate in a more or less mixed ionization mode with the O2+ and NO+ being major impurities with relative proportions to H3O+ typically ranging from 1 to several percent (Amador-Munoz et al., 2016). For instance, for a 1% of the O2+ impurity, the detection limit for hexane would be expected only about 2 orders of magnitude higher than that for a VOC undergoing proton transfer. Therefore, if the n-hexane solvent is used in excess, there is no doubt that high signal will be observed on the charge transfer and hydride abstraction n-hexane ions. I therefore suggest it is clarified how the solvent may have affected the transmission measurements, interferences, and if the n-hexane signal on M86 and M85 and lower alkyl fragments (e.g. M71, M57, M43) may have been saturated.

Noted. The main point here is that provided the CRM is used to constrain the transmission curve then any interference from n-hexane is removed because for the CRMs the solid D3-siloxane is dissolved in the other components, which are all liquids, and so no n-hexane is needed. This point has been clarified by removing reference to the proton affinity in the D3-siloxane preparation section and by adding text in the supporting material (section: Preparation and validation of certified reference materials) to: "**While the proton affinity of *n*-hexane is less than water and therefore does not undergo proton transfer it is still detectable depending on amount fraction because of the presence of minor impurities of $O_2^+$ and $NO^+$ typically 1 – 3 % (Amador-Muñoz et al., 2016), which could cause interferences from signals resulting from the charge transfer and hydride abstract ions. Therefore, an additional advantage of the CRMs is that because the solid D3-siloxane is dissolved in the other components and no n-hexane is used so any potential interferences from the presence of n-hexane are avoided**.". Also, in the main text (section 3.5 CRM Validation) the follow has been added to point out the advantage of using the CRMs in the field as this problem is avoided: "**An additional advantage of the CRMs is that because the solid D3-siloxane is dissolved in the other components no *n*-hexane is used which avoids any potential interferences from the presence of reagent ions other than $H_3O^+$ like $O_2^+$ and $NO^+$.**"

**R2.6** L221 It seems greatly overemphasized that D3 is challenging because of its low vapor pressure. It may be counterintuitive but despite D3 being solid at the room temperature, its vapor pressure is actually high (11.6 mmHg at 25 C) and has a low boiling point of 131 C +/-8C at atmospheric pressure. It means that the D3 solid is unique and readily sublimes. I suggest changing "because of low vapor pressures" to "because of its unique phase transition properties" or something along those lines. I would also suggest to include some relevant properties such as boiling points and vapor pressures to the table.

Added the following text to clarify this: "The final group is comprised of D3-siloxane and 1,2,4-TCB where the spread in validation data is within 10 % and these compounds represent those which the

most challenging to prepare as a result of **either unique phase transition properties** or low vapour pressures, **respectively.**" We have added a table in the Supporting Information (Table S2) that includes physical properties (boiling points and vapour pressures).

**R2.7** TMB has a lower vapor pressure than D3 and D4. I do not think it is critical but it fits better the 3rd category. A table with vapor pressures and boiling points could be useful.

The groupings were based more on the agreement in the validation data. 1,2,4-TMB while it has a lower vapour pressure than D3- and D4-siloxanes, we have a great deal more experience in handling this chemical as it is used in our 30 component ozone precursor mixture that we have been preparing for more than 15 years. We have added a table (Table S2) of boiling points and vapour pressures to the supporting information.

**R2.8** Figure 3 top right panel looks exactly as I would expect an outstanding standard to behave. However, I wonder about the reason for an unexpected slight instability of other compounds presented in the other panels. For instance, why is D3 (and acetone in SI Fig. S2) being generated over time and why are other compounds depleted if there are no oxidants in this relatively high concentration standard (1 ppm) and given the unique proprietary passivation of the cylinder that was promising a longer stability compared to a regular standard.

You need to be careful not to over interpret the trend data because they are based on multiple reference materials as any differences between them (as evidence from the validation data) may influence these as well as changes to the instrument over the long time periods involved. Differences between detector response between the different stability measurement points assumes that every compound behaves like benzene (the well behaved reference) but this is unlikely to be the case and slight differences will also influence the data and lead to artefacts in the data that may result in very small temporal trends. To address this it would be necessary to follow individual mixtures for longer periods of time. This was not possible in this work but is of interest going forward. The observed changes are within the observed differences observed in the validation work and demonstrate that these compounds are stability with the quoted uncertainties for at least 2 years currently.

**R2.9** What was the regulator and type of surfaces used and could metal surfaces be an explanation for a less excellent stability of the compounds? Methanol stabilization on metal surfaces is a known issue that should not be confused with the excellent preparation of the standard. Only two years of stability is decent but maybe slightly less than absolutely outstanding and it would be nice to improve that aspect if not for this mixture maybe in the future.

We use only silconert-2000 (sulfinert) coated tubing and fittings (Restek Corporation) and use a custom minimum dead volume connector based on a variable restriction of a flattened 1/16" tubing to control the pressure drop and for flow control instead of a regulator, needle valve or mass flow controller. As such there is minimal surface interactions between the gas stream and what surfaces are contacted are passivated with silconert-2000 (sulfinert).

**R2.10** PFTBA should be spelled out on its first use. It is a very interesting compound that would make sense to describe a little further to the curious audience.

Perfluorotributylamine has been spelled out on its first use in the abstract and in the introduction.

**R2.11** Why were D6 and D7 siloxanes unincluded? This is surprising because their vapor pressure is still sufficiently high that can be seen even in the highly diluted atmosphere (e.g. Karl et al., 2018). At least adding D6 should have been feasible.

We had not considered these at this time. Thanks for the suggestion. The composition of the transmission curve standard is not fixed and addition of relevant or important compounds could be considered in the future. We would need to look at the physical properties of these two components to determine whether it is feasible to accurately add them to a high pressure gas cylinder at an amount fraction in the µmol/mol range, which is different to whether it could be present in the gas phase at atmospheric pressure and a lower amount fractions.

**R2.12** It would be appropriate to discuss the effect of compound purity. For example, for a 98% purity, if the vapor pressure of compounds making up that 2% is orders of magnitude higher than the compound making up the 98%, the 2% might completely dominate the PTR-MS signal and potentially interfere with other compounds' protonated ions or fragments. I think it would be useful to show some PTR-MS data if you have analyzed the spectrum of the individual 98% pure compound – or if there is a different way to find out what exactly the impurities were?

I am not sure I really understand the comment that if the vapour pressure of impurities is much higher than the main compound then the impurities may dominate the PTRMS signal. It is the proton affinity and the amount fraction that would dictate the response in the PTRMS. Perhaps the reviewer is thinking about the headspace vapour diffusion method that they mention later. In this work, we have purity analysed all the liquids that we have used for the preparation of the PRMs and CRMs in accordance with ISO19229 and this is the origin of the purity data. We do not rely on the purity specified by the manufacturer. Following ISO19229 we look for critical and significant impurities. Critical impurities are those whose presence which would directly impact the amount fraction of other compounds in the mixture. For example, if methanol is present as an impurity in the ethanol then for the PTRMS mixture the methanol impurity would be critical because methanol is also present in the mixture. In this case the amount fraction of the methanol impurity is added to the amount fraction of the methanol that is directly added to give a total amount fraction of methanol in the mixture. This is done for all components. Significant impurities are those that would impact the uncertainty of the mixture by at least 10 % and in the case of the PTRMS could fragment and potentially interfere with other compounds. No impurities were present at 2 %. The major impurities are present at < 0.5 %, most < 0.1 %. As a result of the very small amounts of the pure materials used to prepare the final PRM or CRM mixtures the amount fraction of any impurities in the final mixtures are very small (nmol/mol) and once they are further diluted prior to introduction into the PTRMS instruments (large dilution ratios 40 – 500 used in Holzinger et al., 2019) these are even smaller (pmol/mol) meaning they have a negligible effect. Interferences are negligible. For example, even for impurities that are structural isomers, e.g., 1,2,3-TMB and 1,2,5-TMB (structural isomers of 1,2,4-TMB, which is present in the PRMs and CRMs) and are isobaric molecules so indistinguishable in PTR-MS change the amount fraction by less than < 0.05 %, which is significantly smaller than the assigned uncertainty of 3 %. This is evidenced by the fact that all major ions that are produced from a gas standard sample in the PTR-MS can actually be attributed to the target compounds in the standard.

**R2.13** For the validation experiments how was the standard diluted for the GC and PTRMS measurements? I am missing the RH, the MFCs (and their materials of the seat and the seal, presumably Viton-free?). Were temperature and RH consistent in all measurements? I wonder if that could shed more light on the mechanism for the annual drifts for methanol, acetonitrile, acetone, and PFTBA.

For the validation experiment the mixtures were analysed directly at 1 umol/mol (ppm) of the GC-FID/MS and cryoGC-FID instrument, they were not diluted and no dilution system was used. The mixtures were not analysed by PTR-MS as part of the validation or stability work. We use only silconert-2000 (sulfinert) coated tubing and fittings (Restek Corporation) and use a custom minimum

dead volume connector based on a variable restriction of a flattened 1/16" tubing to control the pressure drop and for flow control instead of a regulator, needle valve or mass flow controller. As such there is minimal surface interactions between the gas stream and what surfaces are contacted are passivated with silconert-2000 (sulfinert). The measurements were all conducted in a temperature controlled laboratory and relative humidity does not have any effect on the direct GC analysis of high pressure gas standards.

**R2.14** I am not a huge fan of the long and overly specific titles. I wonder if it might be possible to simplify the title just a little bit. Specifically, it might be considered shifting the emphasis in the title from "comparability" to more generally on "improved quantification" which in my opinion could resonate even more broadly.

We have added "quantitation and" to the title before comparability. However, we think its still important to include comparability in the title because that is one of the main focuses of having SI traceability for this standard which is to ensure that measurements are true (accurate) thus ensuring all measurements are on a level playing field and improving the comparability, which is key for assimilating data from different sites and different groups into potential data products to be used by the wider community of researchers. To shorten the title we have abbreviated "proton transfer reaction mass spectrometry" to "PTR-MS", which is commonly done in other PTR-MS publications in the literature.

**R2.15** It would be valuable to add info on how processing of D3-D5 siloxane signals was done in your PTR-MS work. In the provided reference to Holzinger et al., 2019 it was not mentioned how the Si isotopes and the CH4-loss fragments were dealt with to reconstruct the transmission curve as the approach requires to sum up all the ions specific to the analyte. In addition, it is unclear what the proton transfer reaction constants were used for those siloxanes.

We have added the following information to the supporting information.

**Figure S4.** Fragmentation of cyclic siloxanes in PTR-MS

**Calculation of product ion abundances from the outlined reaction mechansim:**

| Compound | Product ions | | | |
|---|---|---|---|---|
| | P1[a] | P2 | P3 | P4 |
| D3-siloxane | m223.064*1.37 | (m225.044– m223.064*0.13) *1.36 | m207.033 * 1.36 | (m209.012 – m207.033 * 0.12) *1.35 |
| D4-siloxane | m297.083*1.52 | (m299.062– m297.083*0.18) *1.51 | m281.052 * 1.50 | (m283.031 – m281.052 * 0.18) *1.49 |
| D5-siloxane | m371.101 *1.69 | (m373.081 – m371.101 *0.24) *1.68 | m355.071 * 1.67 | (m357.050 – m355.071 * 0.23) *1.66 |

[a] Correction for stable isotopes.

These assume that for D3-siloxane (209.012 / 209.030 and 225.044 / 225.061 Th are unresolved), D4-siloxane (299.062 / 299.080 and 283.031 / 283.049 Th are unresolved) and D5-siloxane (373.081 / 373.099 and 357.050 / 357.067 Th are unresolved). Note, that P4 is typically not detected for D3 and D5.

**R2.16** Overall, it was extremely enjoyable to read through this seminal work, but I think the conclusions and take-home messages could be even further expanded. For example, as a community should we invest more in the gas standards that can last for at least 2 years or would a properly designed and SI-traced liquid stock solution for dynamic calibration in the proper cal. box could allow even more thorough calibrations including compounds which are challenging or impossible to prepare in gas standards such as organic acids, and with the formulations that can span monoisotopic masses at least until 1000 Da.

This is an interest point. To address this, we have added the following text to the conclusion of the manuscript: "**In the short term (next 5 years) the implementation of an SI traceable transmission curve reference material, such as the one described in this work, using a method similar to that described in Holzinger et al., 2019 is the most pragmatic approach to directly address improving the accuracy (trueness) of quantitation and comparability between different instruments and users. This reflects the challenges and complications of rapidly developing a universally accepted calibration system based on pure liquids that is SI traceable. The use of a SI traceable reference material to properly constrain the transmission curve provides a readily applicable framework to ensure confidence in temporal and spatial data to support the use of PTR-MSs in a broad range of application areas. The use of the transmission curve reference material approach should be seen as a pre-requisite and a complement to additional future efforts to provide alternative calibration efforts for specific target compounds where uncertainties of better than 30 % are needed. Alternative approaches would certainly be necessary for those compounds that are unsuitable for inclusion in high pressure gas standards possibly as a result of very low vapour pressures or other complicating factors such as chemical compatibility with other compounds.** "

**Technical**

**R2.17** Introduction: "with high sensitivity (pmol mol-1). ". Should be changed to something like "ultralow detection limits" (sensitivity is not the same as detection limit).

Corrected to "low detection limits".

**R2.18** L136 remove space before percent.

Done.

**R2.19** Entire ms: Ensure consistency with spelling of sulfide/sulphide (either sulfide or sulphide).

These have all been corrected to sulfide.

**R2.20** L95 Provide VICI valve model (and if it contained Viton seals that can potentially obfuscate methanol stability).

The body of the valve (model: A3C6WT) is silconert-2000 (sulfinert) coated and the rotor is Valcon T (polyimide/PTFE/carbon composite) so no viton seals.

**R2.21** 319-321 the use of transmission and transfer curves in one sentence can be rather confusing for some readers.

Transfer has been changed to transmission curve throughout the manuscript.

**References**

Amador-Muñoz, O., Misztal, P. K., Weber, R., Worton, D. R., Zhang, H., Drozd, G., and Goldstein, A. H.: Sensitive detection of n-alkanes using a mixed ionization mode proton-transfer-reaction mass spectrometer, Atmos. Meas. Tech., 9, 5315-5329, 10.5194/amt-9-5315-2016, 2016.

Ammann, C., Spirig, C., Neftel, A., Steinbacher, M., Komenda, M., and Schaub, A.: Application of PTR-MS fro measurements of biogenic VOC in a deciduous forest, International Journal of Mass Spectrometry, 239, 87-101, 10.1016/j.ijms.2004.08.012, 2004.

Blake, R. S., Monks, P. S., and Ellis, A. M.: Proton-Transfer Reaction Mass Spectrometry, Chemical Reviews, 109, 861-896, 10.1021/cr800364q, 2009.

de Gouw, J. and Warneke, C.: Measurements of volatile organic compounds in the earths atmosphere using proton-transfer-reaction mass spectrometry, Mass Spectrom. Rev., 26, 223-257, 10.1002/mas.20119, 2007.

de Gouw, J., Warneke, C., Karl, T., Eerdekens, G., van der Veen, C., and Fall, R.: Sensitivity and specificity of atmospheric trace gas detection by proton-transfer-reaction mass spectrometry, International Journal of Mass Spectrometry, 223, 365-382, 10.1016/s1387-3806(02)00926-0, 2003.

Holzinger, R., Lee, A., McKay, M., and Goldstein, A. H.: Seasonal variability of monoterpene emission factors for a ponderosa pine plantation in California, Atmos. Chem. Phys., 6, 1267-1274, 10.5194/acp-6-1267-2006, 2006.

Holzinger, R., Acton, W. J. F., Bloss, W. J., Breitenlechner, M., Crilley, L. R., Dusanter, S., Gonin, M., Gros, V., Keutsch, F. N., Kiendler-Scharr, A., Kramer, L. J., Krechmer, J. E., Languille, B., Locoge, N., Lopez-Hilfiker, F., Materić, D., Moreno, S., Nemitz, E., Quéléver, L. L. J., Sarda Esteve, R., Sauvage, S., Schallhart, S., Sommariva, R., Tillmann, R., Wedel, S., Worton, D. R., Xu, K., and Zaytsev, A.: Validity and limitations of simple reaction kinetics to calculate concentrations of organic compounds from ion counts in PTR-MS, Atmos. Meas. Tech., 12, 6193-6208, 10.5194/amt-12-6193-2019, 2019.

Müller, M., Mikoviny, T., and Wisthaler, A.: Detector aging induced mass discrimination and non-linearity effects in PTR-ToF-MS, International Journal of Mass Spectrometry, 365-366, 93-97, https://doi.org/10.1016/j.ijms.2013.12.008, 2014.

Steinbacher, M., Dommen, J., Ammann, C., Spirig, C., Neftel, A., and Prevot, A. S. H.: Performance characteristics of a proton-transfer-reaction mass spectrometer (PTR-MS) derived from laboratory and field measurements, International Journal of Mass Spectrometry, 239, 117-128, https://doi.org/10.1016/j.ijms.2004.07.015, 2004.

Taipale, R., Ruuskanen, T. M., Rinne, J., Kajos, M. K., Hakola, H., Pohja, T., and Kulmala, M.: Technical Note: Quantitative long-term measurements of VOC concentrations by PTR-MS – measurement, calibration, and volume mixing ratio calculation methods, Atmos. Chem. Phys., 8, 6681-6698, 10.5194/acp-8-6681-2008, 2008.

Yuan, B., Koss, A. R., Warneke, C., Coggon, M., Sekimoto, K., and de Gouw, J. A.: Proton-Transfer-Reaction Mass Spectrometry: Applications in Atmospheric Sciences, Chemical Reviews, 117, 13187-13229, 10.1021/acs.chemrev.7b00325, 2017.